# Advances in Microbial Exopolysaccharides: Present and Future Applications

**DOI:** 10.3390/biom14091162

**Published:** 2024-09-16

**Authors:** Huu-Thanh Nguyen, Thuy-Trang Pham, Phu-Tho Nguyen, Hélène Le-Buanec, Holy N. Rabetafika, Hary L. Razafindralambo

**Affiliations:** 1Department of Biotechnology, An Giang University, Vietnam National University, 18 Ung Van Khiem, Long Xuyen City 880000, Vietnam; nhthanh@agu.edu.vn (H.-T.N.);; 2Vietnam National University Ho Chi Minh, Thu Duc City, HCM City 71308, Vietnam; 3INSERM U976-HIPI Hôpital Saint Louis, 1 Avenue Claude Vellefaux, 75010 Paris, France; helene.le-buanec@inserm.fr; 4ProBioLab, 5004 Namur, Belgium; probiolab@europe.com; 5TERRA Research Centre, Gembloux Agro-Bio Tech, University of Liege, Avenue de la Faculté 2B, 5030 Gembloux, Belgium

**Keywords:** exopolysaccharides, probiotics, lactic acid bacteria, immunobiotics, pharmaceutical applications

## Abstract

Microbial exopolysaccharides (EPSs) are receiving growing interest today, owing to their diversity in chemical structure and source, multiple functions, and immense potential applications in many food and non-food industries. Their health-promoting benefits for humans deserve particular attention because of their various biological activities and physiological functions. The aim of this paper is to provide a comprehensive review of microbial EPSs, covering (1) their chemical and biochemical diversity, including composition, biosynthesis, and bacterial sources belonging mainly to lactic acid bacteria (LAB) or probiotics; (2) their technological and analytical aspects, especially their production mode and characterization; (3) their biological and physiological aspects based on their activities and functions; and (4) their current and future uses in medical and pharmaceutical fields, particularly for their prebiotic, anticancer, and immunobiotic properties, as well as their applications in other industrial and agricultural sectors.

## 1. Introduction

Microbial exopolysaccharides (EPSs) are extracellular carbohydrate-based biopolymers produced by various kinds of microorganisms such as bacteria, fungi, yeasts, and microalgae [1,2,3,4]. Bacterial EPSs are synthesized through different pathways and then secreted, either in bound form around the cell surface (capsular EPSs) or in free form into the medium of cell growth (slime EPSs). Once released in the surrounding medium, EPSs associate with other biomolecules like proteins, lipids, and uronic acid derivatives to form an extracellular matrix [5]. Consequently, the term EPS has been used to characterize polysaccharides that represent approximately 40 to 95% of the extracellular polymeric substances secreted by microorganism into the surrounding environment [6]. According to their monosaccharide content, EPSs are mainly classified as homopolysaccharides (HoPSs), that is, polysaccharides with one kind of sugar unit (e.g., glucose or fructose), or heteropolysaccharides (HePSs), those containing different types of monomer residues (e.g., D-glucose, D-galactose, and L-rhamnose), and may include non-carbohydrate groups (e.g., phosphate, acetyl, or amine groups). EPSs also vary in molecular mass (~0.5–2.0 × 10^6^ mol/g), size, sugar linkage pattern, branching degree, and neutral or ionic functional groups [7]. 

Owing to the diversity in chemical structure, various microbial sources, and multiple properties and functions, as well as the environmental compatibility and non-toxicity, EPSs find a wide range of applications in many areas of food, including dairy products and beverages [7,8,9], and non-food areas such as cosmetics, textiles, environmental, agricultural, medical, and pharmaceutical industries [10,11,12]. 

Among EPS-producing microorganisms that have recently received special attention from scientific and industrial communities are LABs [7,8,12]. In fact, more than 40 species of LAB, i.e., microorganisms being generally recognized as safe (GRAS), or having the Qualified Presumption of Safety (QPS), have been reported to produce a wide range of EPSs without health risk [7]. While currently used for their hydrocolloid techno-functionalities as thickeners, stabilizers, emulsifiers, and gelling and water-binding agents in fermented food (e.g., yoghurt and cheese), and pharmaceutical product formulations (e.g., excipients, drug delivery agents), microbial EPSs have also been reported to provide numerous health-promoting benefits for humans, owing to their various biological activities and physiological functions [5,10,11,12]. Antioxidant, antihypertension, hypocholesterolemic, antiviral, antitumor, and immunomodulating activities are some examples currently reported in the literature [7,10,12]. More emergent and relevant functions of EPSs from LABs are their potential use as immunobiotic adjuvants and smart drug delivery systems in vaccine preparation and theranostic design, respectively, which should deserve much more particular attention now and in the future [11,13]. 

The goal of the present paper is to review the diversity of microbial EPSs regarding (1) chemical and biochemical aspects, including composition, biosynthesis, and bacterial sources belonging mainly to LABs or probiotics; (2) technological and analytical aspects, especially their production mode and characterization; (3) biological and physiological aspects based on their activities and functions; and (4) the present and future medical and pharmaceutical applications in connection to their prebiotic, antioxidant, anticancer, and immunomodulation activities against intestinal infections, by particularly emphasizing their role as immunobiotics.

## 2. Microbial EPS

### 2.1. Chemical Structure, Composition, and Types of EPS

EPS composition and chemical structure have been widely investigated by different analytical techniques in terms of monosaccharide units, molecular mass and size, linkage between monomers (glycoside bounds), functional groups (ionic or non-ionic EPSs), branching structure, and microstructure [14,15,16]. Table 1 summarizes different variants and classification criteria in the microbial EPS molecular structure, with some examples of producing microorganisms.

Their classification and name are attributed as a function of these chemical structure criteria. Based on the EPS primary structure, i.e., the monosaccharide composition, it is possible first to distinguish HoPSs from HePSs, that is, EPSs having only one or more type of sugar units. Second, the linkages between units and the position of the carbon involved in the bond C1, C3-α/β, C1, C4- α/β, or C1, C6- α/β, named glycosidic linkages, allow the sub-classification of EPSs. Third, the neutral or ionic nature of functional groups and the degree of chain branching are other sub-criteria of EPS categories within HoPSs and HePSs. Furthermore, uncommon variants such as the oligomeric repeating with the same units (e.g., pentameric of galactose), with three to eight monosaccharide derivative different units (HePSs), or with an approximately equal proportion of glucose and galactose subunits (e.g., kefiran), are among the structural criteria of bacterial EPSs [17]. Some examples of EPS chemical structure are shown in Figure 1.

### 2.2. Biosynthesis and Producing Microorganisms

#### 2.2.1. Biosynthesis of EPS

Despite the wide variety of EPS chemical structures, microorganisms produce EPSs via four different pathways: the Wzx/Wzy-dependent pathway, the ATP-binding cassette (ABC) transporter pathway, the synthase-dependent pathway, and extracellular synthesis using a single glycosyltransferase (Figure 2). In the same species of microorganisms, two or more pathways can coexist to produce different macromolecules [18]. 

HoPSs are biosynthesized by specific glycosyltransferase or fructosyltransferase through two reactions [19].These are the sucrose hydrolysis and the transfer glucosyl (or fructosyl) to the glucan (fructan) polymer chain or oligosaccharide synthesis before creating the final EPS. While the glycosyltransferase catalyzes the biosynthesis of glucans (e.g., dextran, mutan, alternane, or reuteran), the fructosyltransferase synthesizes fructans (e.g., levan and inulin) [20].

The so-called Wzx/Wzy-dependent pathway relates to Gram-positive bacteria [18]. Here, individual repeating units are linked to an undecaprenol diphosphate anchor (C55) at the inner membrane, which is assembled by several glycosyltransferases and translocated across the cytoplasmic membrane by flippase, a Wzx protein. In the next step, their polymerization by the Wzy protein occurs in the periplasmic space before they are released to the cell surface. The transport of polymerized repeat units to the cell surface depends on the polysaccharide co-polymerase (PCP) and the outer membrane polysaccharide export (Outer Polysaccharide Export (OPX) or Outer Membrane Auxiliary (OMA)) families. All polysaccharides synthesized by the Wzx/Wzy path display diverse monosaccharide composition, including in their chemical structure up to four or five sugars. They are therefore classified as HePSs (e.g., xanthan) [16]. 

The ABC transporter pathway mainly occurs in Gram-negative bacteria for CPS production [18]. Such polysaccharides do not really represent EPSs since they are still attached to the cell surface. CPS, synthesized through ABC transport-dependent pathway, is assembled by a single glycosyltransferase on the cytoplasmic surface of the inner membrane. This process yields HoPS or HePS when multiple glycosyltransferases are involved for the assembly process [16]. The synthesized polysaccharides are exported through an efflux pump complex, which includes ABC transporters that span the inner membrane, periplasmatic proteins of the PCP and OPX families, and the outer membrane [18]. 

The synthase-dependent pathway is involved in the synthesis of both CPS and EPS. This pathway utilizes a single synthase complex to perform both polymerization and transport, secreting complete polymer chains directly onto the membrane and cell walls [18]. In the absence of membrane anchors, a receptor protein for signaling molecules, such as bis-(3′-5′)-cyclic dimeric guanosine monophosphate (c-di-GMP), may help initiate polysaccharide assembly. In Gram-negative bacteria, polysaccharides are often transported across the outer membrane [21]. The synthase-dependent pathway is commonly used to assemble HoPSs that require only one type of sugar precursor, as seen in biosynthesis of curdlan, cellulose, alginates, and hyaluronic acid [16].

In the biosynthesis of EPSs through the Wzx/Wzy-dependent, ABC transporter-dependent, or synthase-dependent pathways, polysaccharides are synthesized from nucleotide diphosphate sugars [21]. These sugar nucleotides are produced through a multistep process starting with glycolytic intermediates such as glucose-6-phosphate or fructose-6-phosphate. 

Initially, glucose-6-phosphate is converted into glucose-1-phosphate (Glc-1-P) by phosphoglucomutase. Glc-1-P is then utilized to form sugar nucleotides like uridine diphosphate-glucose (UDP-Glc) and thymidine diphosphate-glucose (TDP-Glc) through the action of UDP-Glc pyrophosphorylase and TDP-Glc pyrophosphorylase, respectively. These sugar nucleotides further transform into other forms: UDP-Glc is converted to uridine diphosphate-galactose (UDP-Gal) by UDP-Gal-4-epimerase, or to uridine diphosphate-glucuronic acid (UDP-GlcA) by UDP-Glc dehydrogenase. TDP-Glc is converted into thymidine diphosphate-rhamnose (TDP-Rha) via the action of TDP-Glc dehydratase. Additionally, Glc-1-P can be converted into guanosine diphosphatefructose (GDP-Fruc) through a series of three intermediates—mannose-6-phosphate, mannose-1-phosphate, and GDP-mannose—by the sequential action of phosphomannomutase (PMM), Man-1-P guanylyltransferase, GDP-mannose pyrophosphorylase, and GDP-mannose dehydratase (Figure 3). The formation of these sugar nucleotides (TDP-Rha, UDP-Gal, UDP-GlcA, and GDP-Fruc) is crucial for EPS biosynthesis, as they serve as precursors for the repeating units that contribute to the diversity of EPS structures [15].

#### 2.2.2. Producing Microorganisms 

EPSs can be generated by a diverse range of microorganisms. These compounds are found outside the cell wall, where they may either adhere to cells, forming capsules, or be secreted into the extracellular environment, forming a slime layer [22]. EPSs are produced by various genera of archaea, bacteria, fungi, and algae, with these microorganisms predominantly falling into mesophilic, thermophilic, and halophilic categories [23]. Among these, bacteria, fungi, yeasts, and microalgae are the most common producers [24], as listed in Table 2. 

##### Bacteria

The ability to produce polysaccharides is common among various bacteria, including species from the *Lactobacillus*, *Streptococcus*, *Xanthomonas*, and *Acetobacter* families. Bacteria can produce both HoPSs and HePSs. 

Dextran, a polymer of D-glucose, is primarily produced by several genera of LAB such as *Leuconostoc*, *Streptococcus*, *Weisella*, *Pediococcus*, and *Lactobacillus* [59]. Commercial dextran is biosynthesized by the non-pathogenic bacterium *Leuconostoc mesenteroides* NRRL B-512 [60]. The Food and Drug Administration (FDA) approves dextran for use in food, cosmetic, and medical applications [59]. Another important glucan is curdlan, which is produced by a variety of bacteria such as *Agrobacterium* spp., *Pseudomonas* spp., and *Bacillus* spp. [61]. Curdlan is a linear, insoluble HoPS composed of 400–500 D-glucose residues linked by β-(1–3)-glucosidic linkages [62]. Water-soluble derivatives of curdlan are utilized in various applications, including as immune recognition sites of dectin-1, and in anti-HIV agents to inhibit Human Immunodeficiency Virus (HIV) infection [63]. 

Levan is a natural fructan found in various plants and microorganism species. This HoPS is made up of D-fructofuranosyl residues linked together by β -(2–6) bonds. Levan is synthesized outside the cell and can be produced by fermentation of sucrose by bacteria such as *Zymomonas mobilis* or *B. subtilis* [62,64]. Alternatively, it can be synthesized enzymatically using levansucrase with sucrose as the substrate. Both Gram-positive bacteria, like *Bacillus* species, and Gram-negative bacteria, such as *Z. mobilis*, are known to produce levansucrase [19].

Xanthan is among the most studied HePS. It is synthesized by *Xanthomonas campestris*. Xanthan features a backbone of glucose with side chains made up of a trisaccharide unit consisting of two mannose residues alternating with glucuronic acid [65]. This EPS typically has repeating units of 2–8 monomers, and exhibits a high molecular weight, ranging from 500 to 2000 kDa, depending on the bacterial genus and species [19]. Xanthan is available in various purity grades for use in food, pharmaceutical, and oil recovery industries [24].

##### Fungi and Yeasts

EPS production is widely distributed among fungi such as the members of the genera *Aureobasidium*, *Candida*, and *Cryptococcus*. The most studied fungal origin polysaccharides are pullulan, scleroglucan, and yeast-glucans. In addition, mushroom polysaccharides such as lentinan, ganoderan, and schizophyllan also have high biological and medicinal properties, which have been gaining increasing attention in recent decades. 

Pullulan is produced through the fermentation of black yeast, such as *Aureobasidium pullulans*. It is currently utilized in the food and pharmaceutical industries due to its unique properties [66]. Structurally, pullulan is a (α-1,4)→(α-1,6)-glucan composed of maltotriose units. In each maltotriose unit, three glucose molecules are linked by α-1,4 glycosidic bonds, while consecutive maltotriose units are connected by α-1,6 glycosidic bonds. Pullulan plays a significant role in protecting the fungal cell from desiccation, and aids in the transport of molecules within the cell [67].

Scleroglucan, the largest β-glucan, is a high-molecular-weight, nonionic polysaccharide produced by fungi like *Botrytis cinerea*, *Schizophyllum commune*, *Sclerotium rolfsii*, *Sclerotium glucanicum*, and *Epicoccum nigrum*. Its structure consists of a (1,3)-β-D-glucopyranosyl backbone with (1,6)-β-D-glucopyranosyl branching residues [68]. Scleroglucan and some derivatives are used in pharmaceutical applications, and in particular for the formulation of modified-release dosage forms [69].

Mushroom polysaccharides include a large group of fungal polysaccharides. Lentinan, a β-(1,3) glucan with β-(1,6)-D-glucose side chains (branching at every third main chain unit), is one of the most important mushroom polysaccharides produced by *Lentinus edodes* [70]. Another example is schizophyllan, an extracellular β-(1,3), β-(1,6) glucan from the filamentous fungus *Schizophyllum commune*. Schizophyllan shares a similar structure with lentinan regarding the side chains and the branching frequency. They also exhibit comparable immunomodulatory and anticancer properties [71]. Ganoderan from *Ganoderma lucidum*, along with grifolan from *Grifola frondosa* and pleuran from *Pleurotus ostreatus*, are other β-(1,3), β-(1,6)-branched mushroom glucans known for their immunomodulatory properties [72]. Additionally, *Ganoderma lucidum* produces a similar immunostimulating endopolysaccharide in submerged cultures [73]. Other medicinal mushroom polysaccharides include krestin (PSK), a HePS proteoglucan from *Trametes versicolor* [74]. *Agaricus blazei* and *Agaricus brasiliensis* produce various EPSs such as glucans and proteoglycans which are safe as functional foods in managing obesity and diabetes [75]. Proteoglycans derived from Cordyceps species (*C. militaris* and *C. sinensis*), which contain glucose, galactose, arabinose, and several amino acid residues, have been investigated for their immunomostimulating and hypolipidemic properties [61,76]. Additionally, research is ongoing into the biological activities of other mushroom polysaccharides [77]. 

Cell-wall β-glucans from *S. cerevisiae* (Y-BG), commercially known as ZymoSan or Zymocel, feature a main chain of β-(1,3)-glucose with β-(1,6)-glucose branches. This immunomodulating and prophylactic β-glucan (proteoglucan) may also include mannans and amino acids. It has been approved for use in the EU by the European Food Safety Authority (EFSA) and in the USA by the FDA [72]. The commercial yeast β-glucan Betafectin (Poly-[1-6]-D-glucopyranosyl-[1-3]-D-glucopyranose) modulates the gut microbiota, promoting the growth of beneficial probiotic bacteria, which in turn produce immunostimulating agents such as short-chain fatty acids (SCFAs) [78].

##### Microalgae and Cyanobacteria

EPSs produced by microalgae exhibit complex chemical structures, ranging from HoPSs, which contain glucose or galactose, to HePSs, which include a variety of different sugar monomers. EPSs from microalgae present a high diversity of sugar monomers. The presence of rare sugars (such as fucose, rhamnose, and ribose), along with uronic acids and sulfates, is common in microalgal EPSs, contributing to their unique properties.

*Porphyridium* sp. is one of the microalgae exploited for commercial EPS production. Common microalgae species known for producing EPS include *Arthrospira platensis*, *Aphanizomenon*, *Chlorella vulgaris*, *Dunaliella salina*, and *Porphyridium cruentum* [24]. *Spirulina platensis* produces sulfated polysaccharides, such as Calcium spinilan, which inhibit tumor invasion and metastasis [79]. 

EPSs produced by cyanobacteria are primarily composed of high-molecular-weight HePSs. Cyanobacterial EPSs can be classified into two main types: those associated with the cell surface (e.g., slime, sheath, and capsules) and those released into the surrounding medium, referred to as released polysaccharides (RPSs) [80]. It is widely acknowledged that the majority of the cyanobacterial EPSs are composed of CPSs, with very few identified as RPSs [81].

Cyanobacterial EPSs have unique characteristics due to the presence of uronic acids and sulfate groups, which confer a negative charge, classifying them as anionic polysaccharides. Compared to bacterial EPSs, cyanobacterial EPSs exhibit a higher degree of monosaccharide diversity, typically containing six or more different monosaccharides. Glucose, xylose, arabinose, galactose, and fucose are the most commonly monosaccharides constituting cyanobacterial EPSs [82]. Methyl sugars and amino sugars have also been reported in the chemical structure of cyanobacterial EPSs [83]. In addition to sulfate groups, which are unique to archaea and eukaryotes, other possible groups include succinyl, pyruvyl, and methyl residues [84]. Common cyanobacteria species that produce EPSs include *Nostoc* spp., *Anabaena* spp., *Phormidium* spp., and *Microcystis* spp. [48]. 

There is growing interest in the large-scale production of cyanobacterial EPSs due to their potential industrial applications such as their use as gums, bio-flocculants, soil conditioners, and biosorbents. Spirulan, Immulan, Nostoflan, and Emulcyan are some examples of commercially available cyanobacterial EPS produced by *Arthrospira platensis*, *Aphanotece halophytica*, *Nostoc flagelliforme*, and *Phormidium*, respectively [84]. Compared to other Gram-negative bacteria, cyanobacteria possess a more significant, thicker peptidoglycan layer, with a greater degree of crosslinking between polysaccharide chains. These unique features of cyanobacterial cell walls make them particularly effective in removing heavy metal from wastewater and outperforming other Gram-negative bacteria. For instance, significant heavy metal removal has been observed with EPSs from *Nostoc muscorum* and *Cyanothece* sp. CCY 0110 [85,86]. 

### 2.3. Production and Analytical Characterization

#### 2.3.1. Production

The process of producing microbial EPSs varies, depending on the species and growth conditions. Environmental changes influence microorganisms, thereby affecting enzyme activity (inhibition or stimulation), protein synthesis (induction and repression), and cell shape. For example, bacteria can produce EPSs in concentrations ranging from 0.29 to 100 g/L in a short time (0.5–7 days), compared to fungi, which have longer cultivation durations (2–32 days) [24]. Controlling cultivation factors such as pH, temperature, dissolved oxygen concentration, aeration, and mixing and stirrer speed is crucial for achieving reproductible bioprocess performance [24]. 

At the lab scale, the cultivation mode is determined by whether EPS production is linked to microbial growth, as with gellan, or occurs independently of growth, as seen with curdlan [87,88]. Most microbial EPS production processes utilize either simple batch cultures or single-pulse fed-batch cultures, typically after the nitrogen source in the medium has been depleted [81]. Other cultivation methods, such as fed batch and continuous culture, have also been proposed. Continuous culture systems are generally more productive; however, there is a greater risk of contamination and they may lead to the development of genetic variants with lower yields [89]. 

At both lab and industrial scales, the synthesis of EPSs is often conducted in stirred tank bioreactors (STRs) or air lift bioreactors (ALRs) [90,91]. The packed bed bioreactor is another type of fermenter configuration [92]. The advantages of STRs are good mixing properties and volumetric productivities, whereas ALRs have the advantage of reduced energy input with efficient heat transfer [93]. 

The method chosen for recovering EPSs from the cultivation broth depends on the characteristics of the producing organisms, the type of polysaccharide involved, and the desired level of purity [24]. The downstream processing involves multiple steps: first, cell removal through centrifugation or filtration, followed by the recovery of the polymer from the cell-free supernatant. The polymer is then precipitated by adding a water-miscible non-polar solvent such as acetone, ethanol, or isopropanol. The resulting precipitate can be separated from the solvent–water mixture and dried. Additional procedures such as re-precipitation with diluted aqueous solutions, deproteinization by chemical or enzymatic methods, and membrane processes can be employed to further remove contaminants [24,94,95]. The general scheme of EPS downstream processing is shown in Figure 4. 

Notably, microbial EPSs were never able to find a suitable place in the polymer market due to their very high production costs. The cost-effectiveness of manufacturing microbial EPSs is influenced by various factors, including production volume, the final product’s value, and the availability of affordable raw materials. EPSs have been evaluated for numerous applications, with their feasibility often dependent on whether fermentation offers a more cost-effective solution than chemical synthesis. For small-scale or waste-utilizing processes, solid-state fermentation can be economical, though it faces scalability challenges. In contrast, submerged and continuous fermentation methods, while offering higher yields, require more energy and investment. Additionally, the required downstream processing, especially to obtain high-purity EPSs, can be expensive and labor-intensive when conducted concurrently. 

To improve the industrial-scale production of microbial biopolymers, it is recommended to optimize fermentation conditions, leverage biotechnological tools like genetic and metabolic engineering, and investigate cost-effective fermentation substrates [96]. Furthermore, using complex media for growth is economically impractical due to the high cost of ingredients such as yeast extract, peptone, and salts, which are needed in large amounts. An alternative is to use agricultural waste as a raw material for EPS production, which can lower manufacturing costs. This method is also environmentally beneficial, as it helps manage waste that would otherwise incur disposal costs and potentially cause environmental issues if sent to landfills or soil. By utilizing agro-industrial wastes as substrates, the process becomes more cost-effective and reduces reliance on nonrenewable resources while minimizing the environmental impact of industrial activities.

#### 2.3.2. Purification and Structural Identification

A series of analytical tools have been used for the purification, structural identification, and quantification of bacterial EPS [5]. In summary, the first step begins by its recovery or extraction from the culture media, generally by precipitation. Once more concentrated substrate is available, the following step is to further purify the EPS-based compounds to precisely determine its composition and chemical structure. A quantitative analysis is also performed to determine the yield of the microorganism in producing EPSs. Further qualitative analyses such as the microstructure and surface property determination may complete its structural characterization. Table 3 summarizes the different techniques frequently used, as well as their functions in the purification and structural characterization of bacterial EPS.

## 3. Properties and Functions

### 3.1. Physiological Functions

In bacterial cells, EPSs play an important role in the formation and maintenance of complex microbial communities, such as flocs and biofilms [99]. They participate in the composition of the biofilm layer surrounding microorganism cells. For instance, glycocalyx is a component of EPSs that is essential for the formation of biofilms. EPSs influence the stability of biofilms by mediating interactions between polysaccharide chains [99]. EPSs also enhance cell adhesion to solid surfaces, including the intestinal mucosa [100]. 

In addition to its role in adhesion, biofilm formation plays a crucial role in the adaptation of bacteria to various physical and chemical conditions in their environment. Microorganisms produce EPSs in response to both biotic stress (e.g., competition) and abiotic stress factors (e.g., changes temperature, light intensity, pH, and salinity). This production is also a strategy of adaptation to an extreme environment, as seen in acidophilic or thermophilic species. EPSs form a protective polymer layer around microbial cells, particularly in harsh conditions. Extreme environmental conditions have stimulated microorganisms to develop various adaptive strategies to counteract the adverse effects of extreme temperatures, high salt concentrations, high and low pH, and radiation. Among these strategies, EPS biosynthesis is one of the most protective mechanisms [101]. The high-water content of the polysaccharide layer enhances resistance to osmotic stress, while its anionic properties help capture essential minerals and nutrients. EPSs can assist in metal degradation due to their chelating properties [19]. The polysaccharide envelope also regulates the diffusion of molecules between extracellular and intracellular media. This diffusion activity can help some bacteria resist surfactants and antibiotics [102,103].

The physiological functions of microbial polysaccharides are extremely diverse and depend on their monosaccharide component and structure. EPSs may contribute to human health (Figure 5) via their prebiotic, anticancer, anti-ulcer, immunomodulatory, or cholesterol-lowering effects [104]. 

On the other hand, EPSs represent a category of potential bioactive polymers with health-promoting functions such as anticancer, antioxidant, antimicrobial, antibiofilm, antihypertensive, antiulcer, hypocholesterolemia, and immunomodulatory activities [7,8,10]. Another category of application is the potential use of EPSs as therapeutic agent carriers (drug, probiotic, etc.) and delivery nano- and microsystems [11]. They play roles as both a protectant and a controlled release system, owing to their various physicochemical and functional properties, such as their adhesion and water-binding capacities, and hydrogel-forming properties, while being biodegradable and safe. One promising application of EPSs is their use mainly under derivative forms as antigen-carriers or adjuvant systems in vaccine preparations [105], gene delivery vectors in gene therapy [106,107], and encapsulation agents for improving cell stability and viability, for instance, in the case of alginate beads for encapsulating fibroblast cells [108] and probiotic bacterium strains [109]. 

EPSs have a positive impact on gut microbiota. The most well-known mechanisms by which EPSs interact with the gut microbiota are linked to their prebiotic effects and their ability to inhibit microbial pathogens, thus helping to regulate the microbiota. Gut microbiota ferment EPSs, producing SCFAs, which lower pH levels and promote the growth and diversity of gut microbial taxa [110,111]. The monosaccharides released from the degradation of EPSs influence the composition of the microbiota through cross-feeding interactions. For instance, EPSs from *Bifidobacterium longum* E44 and *Bifidobacterium animalis* subsp. *lactis* R1 can alter the metabolism of *Bacteroides fragilis* when it grows in their presence [112]. Additionally, EPSs from *Bifidobacterium breve* UCC2003 exhibit antagonistic effects by protecting the host against pathogens, indicating a role for EPSs in providing the health benefits typically linked to probiotic strains and in modulating the immune system [113]. Moreover, EPSs can inhibit pathogenic bacteria, promote probiotics (good bacteria), and maintain the balance of intestinal microflora, due to their viscosity and rheological properties [114,115]. 

EPS exhibit antimicrobial properties with resistance to both Gram-positive and Gram-negative pathogens. EPS-Ca6 produced by *Lactobacillus* sp. Ca6 indicates significant antibacterial activity against pathogenic bacteria such as *Salmonella enterica* ATCC 43972 and *Micrococcus luteus* [116]. EPS-DN1 from *L. kefiranofaciens* DN1 has an inhibitory effect on *Listeria monocytogenes* and *Salmonella enteritidis*. Such inhibition increases with increasing EPS concentration [117]. EPSs can inhibit the growth of pathogens by (a) increasing their competition inhibition against pathogenic bacteria in hosts; (b) combining with signaling molecules related to biofilms or glycocalyx receptors in the pathogen surface that hinder the formation of biofilms; or (c) disrupting membrane integrity and loss of soluble proteins [118].

EPSs are also directly or indirectly related to lowering cholesterol [118]. Those produced by *Enterococcus faecium* K1 and *Lactoplantibacillus plantarum* BR2 have been shown to lower the cholesterol level (48.81%) compared to a negative control [119]. Similarly, in in vitro tests, a 45% reduction in cholesterol was achieved by EPS from *L. plantarum* BR2 [120]. Several hypotheses about the cholesterol-lowering mechanism via EPSs have been proposed based on in vitro and animal experiments. These include bile removal, cholesterol assimilation and conversion, co-precipitation effects (between hydrolyzed bile salts and cholesterol), and the promotion of short-chain fatty acid production to lower cholesterol [121,122].

The strong anticoagulant activity of EPS–sulfate derivatives has been demonstrated [123,124]. Heparin Cofactor II (HC II) is a strong inhibitor of thrombin in blood clotting pathways, and these EPS derivatives may interact with HC II to express anticoagulant activity. The sulfated EPS provides an acidic environment and facilitates the inhibitory effects of HC II on thrombin. The sulfated regions and stereochemistry of EPSs can activate HC II through an allosteric mechanism [118].

Microbial EPSs are also capable of performing very good antioxidant activity. The EPS from *Lactobacillus gasseri* FR4 shows good free radical activity and hydroxyl and superoxide radical capture activities, depending on EPS concentration [125]. The antioxidant mechanism of EPSs is due to the hydrolysis of these biological molecules when exposed to acid, due to active hydroxyl hemiacetates. These active substances provide electrons to free radicals, which turn into stable forms, and eventually reduce the concentration of free radicals [118]. In addition, EPS increases the activity of superoxide effutase, serum catalase, and hepatic glutathione S-transferase in vivo, and reduces the serum malondialdehyde concentration and the activity of monoamine oxidase, showing excellent antioxidant and antiaging effect evidences [126,127]. 

Some EPSs also exhibit a strong immune response and show great potential for fighting inflammation and tumors. Dextran may increase the expression of interferon-1 and interferon-γ in salmon kidneys [128]. A high dose of EPS333 (a HePS) isolated from *Streptococcus thermophilus* was shown to stimulate macrophages to release nitric oxide (NO) and increase cellular immune response [129]. The antitumor activity of EPSs is based on immunomodulation, not only in indirect ways, but also by a direct killing effect on tumor cells. Some EPSs may indirectly activate macrophages to enhance their phagocytic capabilities by (a) facilitating the secretion of pro-inflammatory factors (e.g., IL-1 IL-6, and IL-12), as well as interferon (INF-λ), (b) inhibiting the production of anti-inflammatory factors (e.g., IL-10), and (c) ultimately stimulating interaction between immune and tumor cells [129]. For direct killing effects, it has been shown that EPSs can significantly inhibit the proliferation of HepG-2 and BGC-823 tumor cells, especially HT-29 tumor cells [130]. 

Different antiviral properties of EPSs, including high antiretroviral activity (anti-acquired immunodeficiency syndrome), have been reported in the literature [12,131]. For instance, EPSs extracted from *L. plantarum* LRCC5310 can effectively control rotavirus infection [132]. The antiviral activity of sulfated EPSs is related to the structure, which interacts with the signaling system, receptors, or enzymes, and to the negatively charged properties of these polymers [118]. Heparan Sulfate (HS), a receptor involved in viral infection, exists on the surface of cells. Meanwhile, sulfated polysaccharide shows a structural similarity with HS, and for this reason, it can inhibit competitive combination between HS and viruses, contributing to protection against other pathogens [133].

The principal biological and health-promoting properties of EPSs are listed in Table 4.

### 3.2. Physico-Chemical Properties and Functionalities

EPSs are natural metabolites produced by bacteria when their environmental conditions become unfavorable or extreme, due to different existing stress conditions. These compounds play essentially protective and adaptation roles with regards to high variations in temperature, osmotic pressure, pH, and radiation, but also against pathogen microorganisms. EPS confer to their producing bacteria various fundamental properties such as the aggregation and adhesion capacity to the surface by the capsule form [7,186], biofilm formation or destruction on the solid substrates [187], the ability to colonize in host tissues [188], uptake of nutriments through emulsifying properties [189], and protection from external system by chelating action (binding activities) of toxic elements such as heavy metals [190]. These properties enable the bacteria producing EPSs to compete with pathogens when the energy and nutritional starvation conditions occur since most microorganisms are unable to grow, and even survive, under extreme environment conditions [5]. In nature, EPSs are biodegradable and harmful while being unusable as a carbon source by their own bacterial producers [5]. 

Their natural and fundamental properties are dependent on their chemical structure, but also on their state after bacterial secretion, either cell surface-bound (capsular EPSs) or cell-free biopolymer (slime EPSs), in the liquid medium. The diversity in chemical structure (composition, branched degree, linkage pattern, charge, etc.), molecular weight (MW), and physical state generates a wide range of EPS techno-functionalities, such as solubility and rheology, which can be exploitable in many industrial applications. With different MWs, the same type of EPS can generate various product viscoelasticity, owing to the change in macromolecular conformation in solution, and therefore the volume occupied by a polymer chain. For instance, dextran having a high MW and stronger viscoelastic properties affects the bread characteristics more positively than that having a low MW [191].

The bacterial capacity to aggregate, adhere, emulsify, and form biofilm on the solid surface is affected by the functional groups of EPSs (e.g., amphiphilic EPSs) surrounding the cell membrane, whereas their thickening, viscosifying, gelling, stabilizing, water-binding, and heavy metal absorbent (chelating) powers arise rather from their hydrocolloid and bulk-related properties, such as the molecular weight, water solubility, ionized state, and branching degree. Such techno-functionalities are valuable in both food and non-food product formulations. Table 5 illustrates some examples of bacterial EPS and their fundamental properties and techno-functionalities.

## 4. Current and Future Applications of EPSs

### 4.1. Applications in Pharmaceutical and Medical Fields 

Microbial EPSs and their derivatives have a wide range and numerous potential and commercial applications as biomaterial components, bioingredients, and bioactive agents in medical and pharmaceutical areas [11]. Their health-promoting values in humans have generated particular attention within scientists and industries because of their various biological activities and physiological functions, besides their biocompatibility, biodegradability, and non-toxicity [5]. Dextran as a blood plasma volume expander for controlling wounds [210], an alginate for tissue-engineering of bio-artificial organs [211] and its sodium derivative as an antacid protector [212], and pullulan as a drug carrier or coating agent [213] are among a few examples of commercialized EPS applications as medical devices and biopharmaceuticals. Moreover, some bacterial and fungal EPSs, particularly those from LABs, are largely used as universal “health bioingredients” for supplements or functional foods, and excipients for biopharmaceutical formulations of tablets, capsules, creams, gels, and suspensions, owing not only to their various techno-functionalities (water binding, viscosifying, thickening, emulsifying, stabilizing, and gelling capacities) but also to their GRAS status [9]. Xanthan, cellulose and derivatives, gellan, and levan are some examples of bio-excipients and health ingredients, among others, which are used as thickeners, suspension stabilizers in pharmaceutical creams, and disintegrating agents in tablets for oral, ophthalmic, and nasal drug formulations [7,11,12]. Illustrative examples of commercialized EPS are provided in Table 6.

Table 7 summarizes the main current commercial applications in medical and pharmaceutical applications of EPSs and their derivatives reported in the literature to date. 

Two main future pharmaceutical applications of microbial EPSs, especially those from LAB, emerge through the recent literature overview. 

#### 4.1.1. EPSs as Immunobiotic Agents

EPSs can act as immunobiotic agents by playing some roles in tolerogenic activities of the producing LAB or probiotics. The immune-modulatory capacity of these immunobiotics is highlighted by our recent knowledge of the regulatory arm of immunity [228,229,230]. Indeed, many LAB strains activate the tolerogenic arm of immune reaction, as opposed to the inflammatory one, induced by pathogenic microbes. These pathogens include *Escherichia coli* O157: H7 (EHEC), *Salmonella*, *Listeria monocytogenes*, Campylobacter, and HIV-1 or Hepatitis B Virus (HBV). It has been shown in vivo that acidic EPSs and neutral EPSs are involved in the modulation of innate antiviral immune response in intestinal epithelium cells [231]. Different mechanisms have been reported in the literature for explaining EPS immunomodulation activities (Table 8). 

Thanks to their tolerogenic activities, some EPSs from LABs find a new potential application for the preparation of antiviral and antibacterial vaccines. It has already been announced in the past that EPSs could be used as novel adjuvant systems by enhancing vaccine-induced protection to target challenging pathogens, such as new pandemic viruses and resistant bacteria [11]. To date, immunobiotic agents including LAB strains and their metabolites like EPSs are used through their biological activities, either as preventive or curative treatments, to control inflammatory pathologies [234,235,236,237], or as an adjuvant for vaccine Human Immunodeficiency Virus type 1 (HIV-1) vaccine [238]. Indeed, it has been shown that a vaccine preparation containing inactivated Simian Immunodeficiency Virus (SIV) as the active principle, and *L. plantarum* (ATCC8014) as the adjuvant, when administered orally in *Rhesus macaques*, protected all these animals from SIV infection [238]. Recently, EPS from *L. casei* has been shown to increase the effectiveness of the foot-and-mouth disease vaccine [239]. Moreover, owing to their intrinsic innocuousness, long-term viability in host organisms, and their immune-modulatory capacity, some strains of EPS-producing LABs have been used as vectors to elicit immune responses against bacterial (e.g., *E. coli* and *H. pylori*) or viral (e.g., *Influenza virus*, SARS-CoV, and HIV) pathogens [240,241,242,243,244,245]. 

Considering the ongoing knowledge of the tolerogenic arm of immune reaction, and the ongoing discovery of new LAB strains and new genus commensal microbes, these data anticipate that immunobiotic agents, including both producing LABS and their metabolites such as EPSs, will constitute new bioactive agents of the preventive and/or therapeutic arsenal to use in medicine.

#### 4.1.2. EPSs as Smart Delivery Systems

Another future promising application of bacterial EPSs in pharmaceutics is in the field of smart drug delivery systems, particularly when both diagnostic and therapy strategies (theranostics) can be combined [246]. For instance, EPS-coated magnetic material nanoparticles have a potential use in theranostics, such as in the case of the super paramagnetic iron oxide nanoparticles with crosslinked dextran coating (CLIO). These nanosystems can be used in photodynamic therapy by irradiating atheroma cells in carotid arteries [247]. 

### 4.2. Other Industrial and Agricultural Applications

In addition to their pharmaceutical and medical applications, EPSs have other industrial interests, driven by their unique physicochemical properties and potential as eco-friendly, sustainable alternatives to chemical-based polymers. These applications include textiles [248], bioplastics [249], and petroleum [250]. Welan gum shows particularly great promise in petroleum engineering, especially in polymer flooding for enhanced oil recovery (EOR) in high-salinity and high-temperature reservoirs. This potential stems from its ability to thicken aqueous solutions and its strong viscosifying properties [251]. Additionally, chemical modifications to this biopolymer can further improve its thermoviscosifying performance, solubility, and resistant to bacterial degradation, making it highly suitable for EOR applications in harsh environments [252].

In sustainable agricultural, EPSs are essential for effectively managing both farming practices and environmental health. They improve soil properties, helping to create a moist environment, aggregate soil particles, and protect plant cells from environmental stress and predators [50,253]. EPS has been examined for its capacity to bind soil particles together, functioning like a glue thanks to its ionic charges and viscous texture [254]. In wastewater treatment, EPSs are employed for their biofilm-forming, emulsifying, flocculating, and coagulating properties, which facilitate the decolorization of pollutants and the bioremediation of heavy metals [255]. Table 9 provides a summary of recent studies on EPS applications in various areas.

Microbial exopolysaccharides (EPSs) have a wide range of applications, but for human use, they must either meet the GRAS standard or have an affordable method to neutralize any toxic components, particularly in environmental applications such as municipal and water treatment. The regulatory approval process for EPS-based products varies as a function of their intended use. Key regulatory organizations that ensure the safety, efficacy, and quality of these products include the FDA in the USA, and EFSA and European Medical Agency (EMA) in the EU [4,263].

Industries in the food, pharmaceutical, and cosmetic sectors must adhere to strict regulatory frameworks that include safety, labeling, and Good Manufacturing Practice (GMP) standards to gain market approval. For instance, when dealing with biomass derived from microalgae, there may be concerns about toxicity due to potential contaminants like toxins, heavy metals, or pathogens. It is crucial to assess the potential for harmful substances when choosing microalgae for food products. To comply with legal and regulatory requirements, microalgae biomass and related products must undergo thorough safety evaluations before they can be marketed [264].

## 5. Conclusions

Overviewing the advances in microbial EPSs appears today to be relevant owing to their high diversity in sources and chemical structures, as well as the wide range and multiple functions of such biopolymers for health benefits. Bacterial, fungi, yeast, and microalgae EPSs find numerous applications in pharmaceutical and biomedical areas as biomaterials (e.g., tissue engineering of bio-artificial organs), bio-therapeutic agents (e.g., antiviral, antioxidant, and anticancer activities), and bio-excipients or “bio-ingredients” (e.g., thickeners, emulsifiers, and stabilizers) for drug formulations. Some EPSs are already produced and commercialized at the industrial scale (e.g., xanthan, dextran, and gellan), whereas others (e.g., kefiran and levan) are still in research and development stages, while having a lot of potential future applications. Those from lactic bacterial species (Lactobacilli, Bifidobacteria, etc.), or probiotics in general terms, are gaining particular attention because of their GRAS and QPS status, which allow them to be easily used in many sectors from a juridical viewpoint. Moreover, the current trends to use natural, biocompatible, non-toxic, and biodegradable active compounds for health prevention (e.g., strengthening our immune system) and therapy (e.g., fighting cancers and viral epidemics) are favorable in research and development advances, as well as for the commercialization of existing and new bioactive polymers from bacterial EPSs. While the present review indicates the significant knowledge and use of bacterial EPSs in biomedical and pharmaceutical areas to date, efforts should be continued to improve their production strategies. In fact, their high production cost is the main limiting factor in the advance of microbial EPS use. However, some compensation may come from the high interest and added values of their promising biological activities, such as immunobiotic adjuvants for preparing anti-microbial vaccines and anti-AIDS drugs in pharmaceutical and biomedical areas. Another future promising application of bacterial EPSs is in the field of smart drug delivery systems, particularly when both diagnostic and therapy strategies (theranostics) can be combined. Microbial EPSs, and particularly LAB EPSs, could have diverse and growing applications for our modern society in the near future. 

Despite their potential, ongoing challenges in production and purification processes may hinder their scalability and commercial viability due to high costs and low yields. Thus, numerous studies aim to address these challenges and develop practical solutions. Among these are the utilization of affordable substrates and the isolation of novel strains. To enhance microbial productivity and optimize the use of extracellular polymeric substances (EPSs) in industrial and medical biotechnology, it is crucial to clarify the relationships between metabolic pathways and biosynthetic mechanisms. To uncover new EPS biosynthesis routes and understand the fundamentals of EPS production, advanced omics technologies—such as genome sequencing, functional genomics, protein structure analysis, and emerging bioinformatics tools—are being utilized. Additionally, ongoing research aims to maximize EPS production by investigating how various process variables influence biosynthesis and by identifying the most effective extraction and purification methods.

## Figures and Tables

**Figure 1 biomolecules-14-01162-f001:**
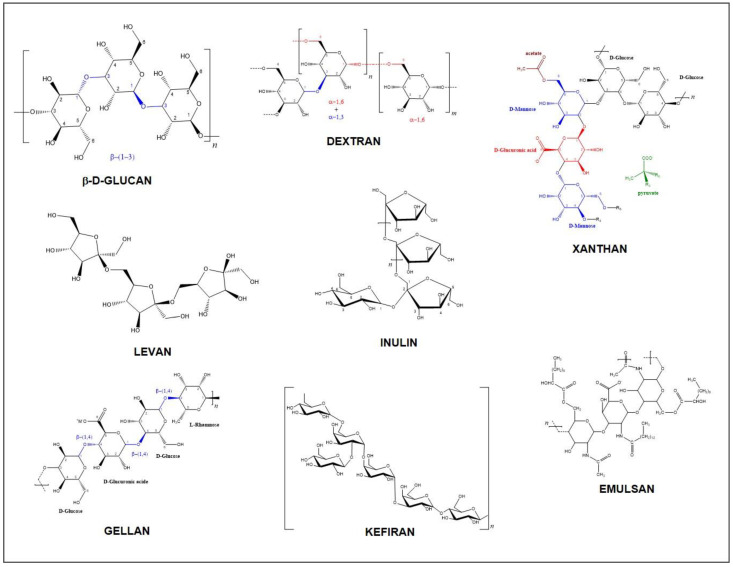
EPS chemical structure examples.

**Figure 2 biomolecules-14-01162-f002:**
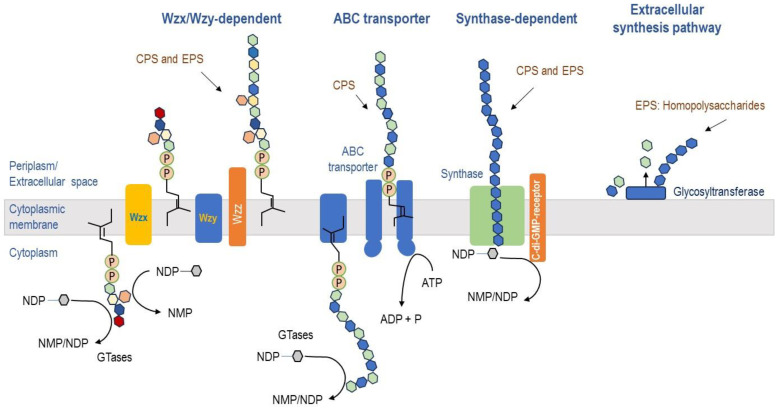
Biosynthesis of polysaccharides in microorganisms. Wzx/Wzy-dependent pathway: Responsible for synthesizing lipopolysaccharide O-antigen polysaccharides in Gram-negative bacteria, as well as capsular polysaccharides (CPSs) and EPSs in both Gram-negative and Gram-positive bacteria. ATP-binding ABC transporter pathway: Facilitates the synthesis of CPS specifically in Gram-negative bacteria. Synthase-dependent pathway: Involved in the synthesis of both CPSs and EPSs in Gram-negative and Gram-positive bacteria. Extracellular synthesis via a single glycosyltransferase: Responsible for the synthesis of EPSs that fall under the category of HoPS.

**Figure 3 biomolecules-14-01162-f003:**
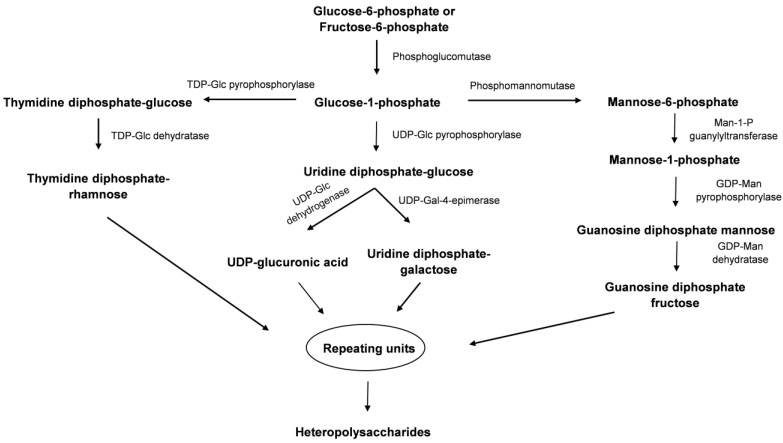
Outline of biosynthesis of HePS. The abbreviations are TDP: thymidine diphosphate; UDP: uridine diphosphate; GDP: guanosine diphosphate; Man-1-P: mannose-1-phosphate; Glc: glucose; Gal: galactose; Man: mannose.

**Figure 4 biomolecules-14-01162-f004:**
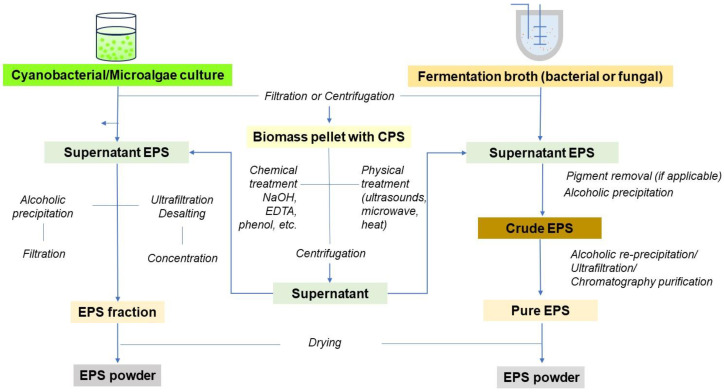
Workflow of EPS production.

**Figure 5 biomolecules-14-01162-f005:**
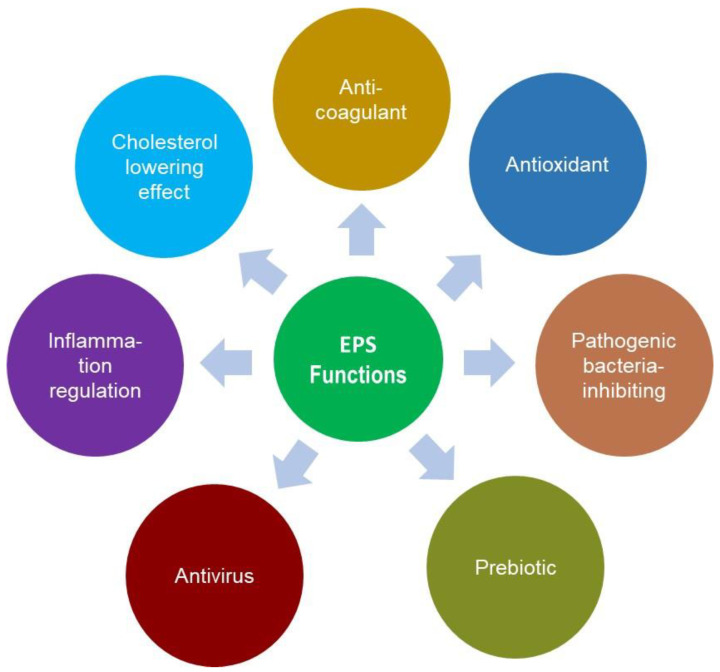
Physiological functions of EPS.

**Table 1 biomolecules-14-01162-t001:** Classification and examples of microbial EPS.

	Monomers	Substituents	Linkage	Branching	Charge	HoPS	HePS
**Bacterial EPS**							
** *LAB EPS* **							
	Glc		β(1→3)	Linear	Neutral	β*-D-glucan*	
			α(1→3)	Linear		*Mutan*	
			α(1→3)	Linear		*Neutran*	
			α(1→6); α(1→4)	Branched		*Reuteran*	
			α(1→6); α(1→3)	Branched		*Dextran*	
			α(1→6); α(1→3)	Linear		*Alternan*	
	Fru		β(2→1)	Linear		*Fructan*	
	Gal			Linear		*Polygalactan*	
	Fru		β(2→6); β(2→1)	Branched		*Levan*	
	Glc; Gal			Branched			*Kefiran*
** *Non-LAB EPS* **							
	Glc				Neutral	*Curdulan*	
	Glc; Man; GlcA	Ace; Pyr	β(1→4)	Branched	Anionic		*Xanthan*
	GulA; ManA	Ace		Branched			*Alginate*
	Glc; Rha; GlcA	Ace; Gly		Linear	Anionic		*Gellan*
		GlcN	β(1→4); β(1→3)	Linear	Anionic		*Hyaluronic acid*
	Man				Neutral		*Xylinan*
	Glc		β(1→4)	Branched	Neutral	*Cellulose*	
**Fungi EPS**							
	Glc		α(1→4); α(1→6)	Linear	Neutral	*Pullulan*	
	Amino sugar	Ace; Gal-N, R-COOH			Anionic		*Emulsan*

Glucose (Glc); fructose (Fru); galactose (Gal); mannose (Man); glucuronic acid (GlcA); guluronic acid (GulA); mannuronic acid (ManA); rhamnose (Rha); glucosamine (GlcN); acetate (Ace); pyruvate (Pyr); glycerate (Gly); galactosamine (Gal-N).

**Table 2 biomolecules-14-01162-t002:** Microbial EPS producers: bacteria, fungi, and microalgae.

Microorganism	Polymer	Sugar Monomers	Non-Sugar Residues	Mw (Da)	Reference
**BACTERIA**					
*Agrobacterium* sp.	Curdlan	Glc	---	5.3 × 10^4^–2.0 × 10^6^	[25]
*Azotobacter vinelandii; Pseudomonas aeruginosa*	Alginate	GulA, ManA	Ace	(0.3–1.3) × 10^6^	[26,27,28]
*Bacillus subtilis*; *Halomonas* sp.; *Zymomonas* sp.	Levan	Fru	---	2 × 10^6^	[29,30]
*Enterobacter A47*	FucoPol	Fuc, Gal, Glc, GlcA	Ace, Pyr, Succ	(1.7–5.8) × 10^6^	[31]
*Klebsiella pneumoniae*	Fucogel	GalA, Fuc, Gal	Ace	4 × 10^4^	[29]
*Acetobacter* sp.; *Glucanoacetobacter* sp.; *Rhizobium* sp.; *Sarcina* sp.	Bacterial cellulose	Glc	---	~10^6^	[29,32,33]
*Lactobacillus* sp.; *Leuconostoc* sp.; *Streptococcus* sp.	Dextran	Glc	---	10^3^–10^7^	[34]
*Pseudomonas oleovorans*	GalactoPol	Gal, Glc, Man, Rha	Ace, Pyr, Succ	(1.0–5.0) × 10^6^	[35,36]
*Sphingomonas paucimobilis*	Gellan	Glc, Rha, GlcA	Gly, Ace	5.2 × 10^5^	[37]
*Streptococcus zooepidemicus*	Hyaluronic acid	GlcNAc, GlcA	---	(2.0–3.0) × 10^3^	[38]
*Xanthomonas* sp.	Xanthan	Glc, Man, GlcA	Pyr, Ace	2.0 × 10^6^–5.0 × 10^7^	[27]
**FUNGI**					
*Tuber borchii*	---	Glc	---	92 × 10^3^	[39]
*Colletotrichum alatae* LCS1	---	Man, Gal, Rha, Ara, Glc, Fuc	---	---	[40]
*Aureobasidium pullulans*	Pullulan	Glc	---	4.8 × 10^4^–2.2 × 10^6^	[2,41]
*Penicillium janthinellum* *N29*	---	Gal, Man	---	10.24 × 10^3^	[42]
*Monascus purpureus*	---	Fuc, Gal, Glc, Man, GalA, GlcA	---	3.2 × 10^5^	[43]
*Penicillium citrinum*	---	Ara, Gal, Glc, Man, GalA, GlcA	---	1.58 × 10^5^	
*Aspergillus versicolor*	---	Ara, Gal, Glc, Man, Xyl, GalA, GlcA	---	1.14 × 10^5^	
*Fusarium merismoides* A6	---	Man, Glc, Gal, Rib	---	(5.14–6.50) × 10^4^	[44]
*Ganoderma lucidum*	---	Gal, Man, Glc, Ara, Rha	---	2.08 × 10^4^	[45]
*Sclerotium* sp.	Scleroglucan	Glc	---	1.3 × 10^5^–6.0 × 10^6^	[46]
*Schizophyllum commune* 227E.32	Schizophyllan	Glc	---	1.1 × 10^6^	[47]
**MICROALGAE**					
*Anabaena augstmalis*	---	Glc, Gal, Man, Xyl, Fuc, Rha, Gal-N, GlcN, GalA, GlcA	Sulf	n.a.	[48]
*Dunaliella tertiolecta*	---	Glc	---	n.a.	[49]
*Scenedesmus acuminatus*	---	Gal, GlcN, Man	---	High (>50 × 10^3^) and low molecular weight (<3 × 10^3^)	[50]
*Phormidium autumnale*	---	Rha, Rib, Man, Glc, Fuc, Gal, Ara, GalA, GlcA	Sulf	n.a.	[48,51]
*Porphyridium sordidum*	---	Fuc, Rha, Ara, Gal, Glc, Xyl, GlcA	Sulf	14 × 10^5^	[52]
*Rhodella* sp.	---	Xyl, Gal, Glc, Rha, Ara, GlcA	Sulf	n.a.	[53]
*Synechocystis aquatilis*	---	Fuc, Glc, Rha, Xyl, Man, GlcN, GalA, GlcA	Sulf	n.a.	[48]
**CYANOBACTERIA**					
*Spirulina platensis*	---	Fru, Rha, Rib, Man, Gal, GalA, Glc, Xyl	Sulf, Ca		[54]
*Nostoc* sp.	---	Ara, Glc, Man, Xyl, GlcA	lactyl	214 × 10^3^	[55]
*Anabaena* sp. CCC 745	---	Glc, Rha, GlcA		19.57 × 10^3^ 30.29 × 10^3^	[56]
*Nostoc* cf. *linckia*	---	Glc, Gal, Xyl, Man, GlcA	lactyl	1.31 × 10^5^	[57]
*Gloeocapsa gelatinosa*	---	Glc, Gal, Ara, Fuc, Xyl, Rha, Man, GlcA, GalA		67.2 × 10^3^ 598.3 × 10^3^	[58]

Fucose (Fuc); arabinose (Ara); xylose (Xyl); ribose (Rib); galacturonic acid (GalA); succinate (Succ); phosphate (Phosp); sulfate (Sulf); n.a (not available).

**Table 3 biomolecules-14-01162-t003:** Different techniques frequently used and their functions in qualitative and quantitative EPS analyses.

Recovery/Purification	Functions	Reference
*Heating—Sonication*	Recovery of CPS	[97]
*Precipitation*		
*Dialysis*	Removing simple carbohydrates	
*Ion-exchange chromatography*	Final purification before quantification	
*Size Exclusion Chromatography (SEC)*		
*Preparative Sodium Dodecyl Sulfate—Polyacrylamide Gel (SDS-PAGE)*		
**Qualitative analysis**		
*Ultraviolet (UV) spectroscopy*	Detection of nucleic acids and proteins	[7]
*Fourier Transform—Infrared (FT-IR) spectroscopy*	Detection of functional group; configuration α or β; fingerprints	[5]
*Gel permeation chromatography or SEC-Multi-Angle Light Scattering (MALS)*	Molecular mass detection	
*Gas chromatography coupled to mass spectrometry*	Monosaccharide composition	
*High performance Anion exchange chromatography (HPAEC)*	Linkage and composition	[5]
*Near Magnetic Resonance (NMR) spectroscopy*	linkage pattern	
*Confocal laser scanning microscopy*	Microstructure analysis	[97]
*Scanning electron microscopy*		
*Transmission electron microscopy*		
*Atomic force microscopy*		[7]
*Differential scanning calorimetry (DSC)*	Structural analysis	
*Thermogravimetric analysis (TGA)*		[5]
*X-Ray diffraction (XRD)*		
*Laser light scattering/electrophoretic analysis*	Physico-chemical properties	[98]
**Quantitative analysis**		
*Gravimetrics*		[97]
*Colorimetrics*		
*SEC-Refractive Index (RI)*		
*Near Infrared (NIR) spectroscopy*		

**Table 4 biomolecules-14-01162-t004:** Biological properties and health-promoting effects of EPS.

Biological Properties and Health Benefits	EPS	Source	Reference
*Anticancer activity and Anticancer adjuvant*			
Antitumor activity by the activation of defender cells against cancer cells	β-glucans-based EPS	*Aureobasidium pullulans*	[134]
Anticancer activity against human colon, liver, embryonic kidney, breast cancer cell lines	Levans	*Lactobacili, Bifidobacteria*	[135]
Antiproliferative effect		*Lactobacillus pantheris* TCP102	[136]
Antitumor activity on HepG-2, BGC-823, HT-29 cancerous cells		*L. plantarum* 70810	[137]
Induced cytotoxicity in colon cancer cell lines		*Limosilactobacillus fermentum* YL-11	[138]
Apoptotic, antiangiogenic effects, and autophagy		*Bacillus sonorensis, Rhodococcus pyridinivorans* ZZ47	[139,140]
*Immunomodulatory activity*			
Immunomodulation effects through interaction with macrophage receptors	β-glucans-based EPS		[141]
Immunomodulation by human macrophage activation (cytokine production)	β-glucans	*P. parvulus*	[142]
Modulate the immune system (innate and adaptive response)		LAB	[143]
Suppressors of the immune response		*Lactobacillus confusus* TISTR 1498	[144,145]
Stimulation of antigen presenting cells (e.g., dendritic cells)		*Limosilactobacillus reuteri* L26, *L. reuteri* DSM17938	[146]
Stimulate production of cytokines by macrophages		*Nostoc* sp.	[55]
Maintaining the immune balance in states of inflammation and/or infection		Lactobacilli	[147]
Improving allergic responses and suppressing allergen specific IgE synthesis		*Leuconostoc citreum* L3C1E7	[148]
Suppression the pro-inflammation and promotion of regulatory cytokine		*S. thermophilus, Bacillus licheniformis, Leu.* *mesenteroides*	[149]
Activation of T lymphocytes and monocytes		*S. thermophilus, Bifidobacterium breve*	[150,151]
Evasion of potentially damaging immune responses		*Bifidobacterium breve*	[151]
Restoration of the mucosal barrier		*Lactobacillus helveticus* KLDS1. 8701	[152]
Immunostimulator	Levan (β-2, 6-fructan)	*B. subtilis natto*	[153,154]
*Antiviral effects*			
Antiviral activity on avian influenza and adenovirus	Levans	*B. subtilis (honey)*	[155]
Anti-AIDS	Curdlan	*Agrobacterium sp.*	[156]
Antiviral against human hepatitis B	Curdulan sulfate		[157]
Effects against enveloped viruses	Dextran sulfate	*Leu. mesenteroides* B512F	[158]
Antiviral and antibacterial activities	β-glucans-based EPS	Fungi	[159]
*Cholesterol lowering and anti-hypertensive properties*			
Hypoglycemic and hypolipidemic activities		*Cordyceps militaris.*	[160]
Lowering blood cholesterol		*Lactiplantibacillus paraplantarum* NCCP 962	[161]
Modulation of lipid metabolism		*Lactobacilli*	[162]
Cholesterol-lowering properties and inhibit α-amylase		*L. plantarum RJF4*	[137]
Antihypertensive effects		*S. thermophilus and Lactobacillus bulgaricus*	[163]
*Anti-diabetes type 2 and hypocholesterolemia*			
Anti-diabetes		*Pseudomonas* sp. strain AHG22	[164]
*Antibiofilm agents*			
Ability to repress biofilm		*Lactobacillus helveticus* MB2-1	[165]
Limiting the biofilm formation on medical devices		*L. fermentum, Leu. citreum, Leu.* *Mesenteroides, Leu. Pseudomesenteroides, Ped. pentosaceus*	[166]
Antiadhesive and antibiofilm activities against oral *S. Aureus* strains		Lactobacilli	[167]
Antibiofilm activity		*L. plantarum-12*	[168]
*Anti-ulcer effects*			
Gastro-protective effect		*L. plantarum* E1K2R2	[169]
Inhibition of the adhesion of *H. pylori*		*Lacticaseibacillus paracasei*	[170]
*Antioxidant activities*			
Antioxidant and antiproliferative activities against human gastric	Levan	*Pantoea agglomerans* ZMR7	[171]
Removal of free radicals scavenging activities		*Halomonas elongata*	[172]
Scavenging of reactive oxygen species (ROS) and reduction in lipid peroxidation		*Bacillus velezensis* SN-1	[173]
Inhibition of H_2_O_2_ induced apoptosis		*L. plantarum* C88	[137]
Ferrous ion chelation		*Gloeocapsa gelatinosa*	[58]
*Prebiotic activities*			
Prebiotic effect	HePS	*L. paracasei*	[174]
Bifidogenic effect	Dextran	*Leu. mesenteroides*	[175]
*Pathogen Antagonism*			
Prevent binding of pathogenic bacteria to mucus	HePS	Probiotic LAB	[176]
Reduce the adherence of pathogen to Caco-2 cells surfaces		*L. casei* NA-2	[177]
Inhibit the biofilm formation of a number of pathogens		*B. licheniformis, Leu. mesenteroides*	[175,178,179]
Modulate microbial biofilms	Man-Glu	*Lactobacillus*	[180]
Enhancing intestinal barrier function	Man-Glu-Rib	*L. plantarum*	[181]
*Anti-obesity effects*			
Improved human gut microbiota	Glucan	*Weissella cibaria*	[182]
*Anti-skin irritation*			
Anti-inflammation	Dextran	*Leuconostocaceae*	[183]
*Anti-Alzheimer’s disease*			
Treating Alzheimer’s illness to obviate side effects of synthetic drugs	ManA-Glc-Man-Rha	*Streptomyces*	[184]
Neuroprotective agent against amyloid beta1–42-induced apoptosis in SH-SY5Y cells	Man-Glc-Fru-Ace-Glc-N	*Lactobacillus delbrueckii ssp. bulgaricus* B3	[185]

**Table 5 biomolecules-14-01162-t005:** Bacterial EPS and their properties and techno-functionalities.

EPS	Activities	Producers	Reference
α-D-glucan	Viscosifier	*Levilactobacillus brevis* HDE-9	[192]
	Water holding capacity	*Enterococcus hirae* OL616073	[193]
Levan	Enhancer of Bifidobacteria growth	*Lactobacillus sanfranciscensis*	[194]
Dextran	Enhancer of probiotic growth	*W. cibaria*	[195]
	Promote colonization of strains	*Leuconostoc lactis*	[196]
Kefiran	Enhancer of Bifidobacteria growth	*Lactobacillus kefiranofaciens*	[197]
	Emulsifying and flocculating activities	*L. kefiranofaciens*	[197]
Galactan	Emulsifying capacity and stability; flocculating activity at a wide range of pH	*Weissella confusa*	[198,199]
Anionic EPS	Heavy metal absorption activity (binding agents, bioabsorbents)	Lactobacilli	[200]
[Man:Glc]-EPS	Antibiofilm against pathogens	*L. fermentum* LB-69	[201]
[Gal:Glc:Man]-EPS	Probiofilm	*L. plantarum* KF5	[202]
[Glc:Man 1:7.01] [Man:Glc:Rha 7.45: 1.00: 2.34]	a protector against oxidative stresses or reducing power or agent against free radicals	*L. lactis* subsp. *lactis* IMAU11823	[203]
[Glu:Rib:Man:Xyl 1.0:16.4:6.6:6.5] [Rib:Man:Xyl:GA:Ara 7.1:1.6:4.8:1.0:9.0]	Inhibitor of lipid peroxidation	*Lactobacillus* sp.	[204]
[Uronic acid]-EPS	Protector of cell membrane against lipid peroxidation	*L. plantarum* LP6	[137]
[Gal-Glc-Man]-EPS	Inhibition of H_2_O_2_	*Bacillus sp.* S-1	[205]
[Se]-EPS	Antioxidant activity	*L. lactis*	[206]
[Glc-Man-Gal-Fuc-Glc-NH2]-EPS	Antioxidant effects	*B. coagulans* RK-02	[207]
Gellan	Gelling (flexible elastic gel) and emulsifying activities	*Sphingomonas elodea*	[208]
Pullulan	Coating protector effect against mold	*Aureobasidium pullulans*	[209]

**Table 6 biomolecules-14-01162-t006:** Examples of bacterial EPSs manufactured at the industrial scale.

EPS	Microorganism Producer	Manufacturer	Brand Name	Reference
Xanthan	*Xanthomonas campestris*	CPKelco, San Diego, CA, USA	Keldent	[214]
			Xantural	
Gellan	*Pseudomonas elodea*	CPKelco, San Diego, CA, USA	Gelzan	
			Kelcogel	
Hyaluronic acid	*Streptococcus zooepidemicus* Equi	Contipro Biotech, Dolni Dobrouc, Czech Republic	Sodium hyaluronate	[215]
Pullulan	*Aureabasidium pullulans*	Nagase Viita (Hayashibara Co., Ltd., Okayama, Japan)	Pullulan	[216]
Cellulose	*Gluconacetobacter xylinum*	fzmb GmbH, Research Centre of Medical Technology and Biotechnology	Nanomasque	[217]
		Axcelon Demacare Inc, North York, ON, Canada	Nanoderm	
Dextran	*Leu. mesenteroides*	Meito Sangyo, Nagoya, Japan	Dextran sulfate Na	[218]

**Table 7 biomolecules-14-01162-t007:** Examples of current medical and pharmaceutical applications of EPSs and their derivatives.

Medical and Pharmaceutical Applications	EPS	Source	Reference
Blood plasma volume expander, blood plasma substitute	Dextran	*Leu. mesenteroides*	[219]
Excipients, thickener and suspension stabilizer, drug-controlled release carrier	Xanthan	*Xanthomonas campestris*	[220]
Tablets disintegrant, thickener, emulsion stabilizing agents	Alginate	*Pseudomonas aeruginosa* *Azotobacter vinelandii*	[221,222]
Antiacid protector in capsules	Na-alginate		
Disintegrating agent, drug-controlled release, gelling agent, wound dressing	Gellan	*Sphingomonas elodea* *Sphingomonas paucimobilis*	[223,224]
Binding and film-forming properties, coating agent for oxygen impermeability	Pullulan	*Aurebasidium pullulans*	[225]
Vitreous substitution during eye surgery, intraarticular injections in osteoarthritis	Hyaluronic acid	*Streptococcus equi*	[226,227]

**Table 8 biomolecules-14-01162-t008:** Examples of EPS immunomodulation activity mechanisms.

Mechanism	EPS Producing LAB	References
Interaction with dendritic cells and macrophages	*Lactobacillus bulgaricus*	[232]
Enhancing the proliferation of lymphocytes		
Induction of nitric oxide secretion in vitro	*L. plantarum* JLK0142	[233]
Enhancing the phagocytic potential of macrophages		
Increasing IgA concentrations in the intestinal mucosa		
Stimulating lymphocyte proliferation		

**Table 9 biomolecules-14-01162-t009:** Recent studies of EPS applications.

Uses	EPS	Properties	Reference
Textiles	Xanthan	Improve adhesive properties of textile	[256]
Oil Recovery	Xanthan	Reduce oil spreading	[257]
	Modified xanthan	Enhance oil recovery	[258]
Bioremediation	KO-EPS	Chelate iron	[259]
Agricultural	Xanthan	Improve water absorption and water retention capacity	[260]
	EPS	Antifungal activity	[261]
Packaging	Pullulan	Antifungal activity	[262]

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
