# Peer review of "Advances in Microbial Exopolysaccharides: Present and Future Applications"

_biomolecules, 2024, doi:10.3390/biom14091162_

Round 1

Reviewer 1 Report

Comments and Suggestions for Authors

This is an interesting manuscript exploring microbial EPS and their applications. Overall, the article is well-written and includes necessary information regarding the source, production and some applications of EPS. However, a few concerns need to be addressed, as outlined below.

In the introduction, since the authors mentioned that the goal of the review is to examine EPS produced by probiotics, it would be beneficial to include a section discussing how these EPS interact with the gut microbiome and contribute to the health beneficial properties. This should be discussed in later sections of the manuscript.

Figure 1, while inulin is mentioned, levan, another common microbial EPS, is not included. Please add levan to the figure.

In 2.2.1, the discussion on the four synthesis pathways is unclear regarding their relation to released EPS or CPS. This aspect is vague and needs clarification. It is suggested to revise the section to explain how each pathway relates to EPS or CPS, and reflect this in Figure 2 for better understanding.

In 2.2.2, the manuscript does not mention cyanobacteria, which are significant autotrophic prokaryotes involved in EPS production and have important environmental roles. Please include a dedicated paragraph discussing cyanobacteria as EPS producers.

In Figure 4, it currently only illustrates methods for producing released EPS. It would be helpful to include a flowchart outlining the methods for capsular EPS production, including techniques used for cleaving cell wall-attached polysaccharides.

In 3.2, the discussion on physicochemical properties should include the factor of molecular weight, which is critical for EPS applications. Different molecular weights of the same type of EPS can significantly impact properties such as rheology and solubility. Please address how molecular weight influences these factors.

For the application, the current discussion merely focuses on the potential medical applications of EPS. In fact, EPS has much broader applications across various industries, such as textiles, petroleum, environmental waste treatment, soil conditioning, and even in the development of bioplastics. Including these areas would enhance the scope and impact of the review.

Author Response

1. Summary

2. Point-by-point response to Comments and Suggestions for Authors

Comments 1: In the introduction, since the authors mentioned that the goal of the review is to examine EPS produced by probiotics, it would be beneficial to include a section discussing how these EPS interact with the gut microbiome and contribute to the health beneficial properties. This should be discussed in later sections of the manuscript.

Response 1 Thank you for pointing this out. We agree with this comment. Therefore, we have added information on the EPS interaction with the gut microbiome and its contribution to the health beneficial properties (Line 417-430).

Comments 2: Figure 1, while inulin is mentioned, levan, another common microbial EPS, is not included. Please add levan to the figure.

Response 2: Agree. Fig.1 has been updated with “levan” chemical structure.

Comments 3: In 2.2.1, the discussion on the four synthesis pathways is unclear regarding their relation to released EPS or CPS. This aspect is vague and needs clarification. It is suggested to revise the section to explain how each pathway relates to EPS or CPS and reflect this in Figure 2 for better understanding.

Response 3: We would like to appreciate your comment. The section explaining the fours synthesis pathways has been widely improved. Fig2 has been completed by adding some annotations inside and more explanations to the legend (Line 112 – 118).

Comments 4: In 2.2.2, the manuscript does not mention cyanobacteria, which are significant autotrophic prokaryotes involved in EPS production and have important environmental roles. Please include a dedicated paragraph discussing cyanobacteria as EPS producers

Response 4: We would like to appreciate your comment. We added detailed information on EPS produced by cyanobacteria (Line 273-300).

Comments 5: In Figure 4, it currently only illustrates methods for producing released EPS. It would be helpful to include a flowchart outlining the methods for capsular EPS production, including techniques used for cleaving cell wall-attached polysaccharides.

Response 5: Fig.4 has been updated. We added a flowchart showing the methods for capsular EPS extraction.

Comments 6: In 3.2, the discussion on physicochemical properties should include the factor of molecular weight, which is critical for EPS applications. Different molecular weights of the same type of EPS can significantly impact properties such as rheology and solubility. Please address how molecular weight influences these factors.

Response 6: We would like to appreciate your comment. We added information on the molecular weight impact on EPS properties, functionalities, and applications (Line 506-515).

Comments 7: • For the application, the current discussion merely focuses on the potential medical applications of EPS. In fact, EPS has much broader applications across various industries, such as textiles, petroleum, environmental waste treatment, soil conditioning, and even in the development of bioplastics. Including these areas would enhance the scope and impact of the review.

Response 7: Agree. We added a new section 4.2 dedicated to the other applications of EPS including different industrial and agricultural sectors, which are summarized in Table 9 (Line 597-619).

Reviewer 2 Report

Comments and Suggestions for Authors

Please check the attached review report.

Comments on the Quality of English Language

There are several minor English language mistakes. Please thoroughly check the manuscript for the English language.

Author Response

1. Summary

2. Point-by-point response to Comments and Suggestions for Authors

Comments 1: Large-scale production and commercialization of EPS involve significant economic considerations. The paper should have provided insights into the cost-effectiveness of different production methods and how technological advances could lower costs and improve scalability in the future.

Response 1: We would like to appreciate your comment. We added information on the cost-effectiveness of different production methods and several strategies used to lower costs and improve scalability (Line 334 – 344).

Comments 2: Incorporating insights into more sustainable production methods, including the use of renewable resources and waste minimization, would have significantly enhanced the paper's relevance to modern industrial practices.

Response 2: Thank you for your comment. We added information on using renewable resources and waste minimization in EPS production (Line 344 – 355).

Comments 3: The review should have included a section on the legal framework and approval process for EPS based products, particularly in pharmaceuticals and functional foods. Many reviews in this field overlook this key area, but it is vital for industry adoption.

Response 3: We would like to appreciate your comment. We included the legal framework for EPS-based products (Line 620 – 634).

Comments 4: While the paper explores some future applications, it lacks a broader discussion of research gaps in the field of microbial EPS. The review could benefit from identifying areas where additional research is needed, such as improving yield efficiency, optimizing purification processes, or understanding the interactions of EPS with other compounds in complex formulations.

Response 4: Thank you for your comment. We included several necessary approaches for the study of EPS in the future (Line 663 – 675).

Comments 5: Rewrite the first sentence of abstract “Microbial exopolysaccharides (EPS) are receiving today a growing interest owing to their diversity in chemical structure and source, multiple functions, and immense potential applications in many industries of food and non-food area” as  “Microbial exopolysaccharides (EPS) are receiving a growing interest today owing to their diversity in chemical structure and source, multiple functions, and immense potential applications in many industries of food and non-food area”.

Response 5: The first sentence of abstract has been rewritten and highlighted in the revised manuscript (Line 15-17).

Comments 6: Line 32: change “kind” to “kinds”; Line 76: replace “has” to “have”; Line 88: correct “distinguish HoPs to HePs to “distinguish HoPs from HePs”; Line 188: Replace genus with genera; Line 193: Change Agrobacterium sp., Pseudomonas sp. and Bacillus sp. As Agrobacterium spp., Pseudomonas spp. and, Bacillus spp. If they are multiple species.; Line 215: Change have also to also have; Line 222: change cell protecting to protecting cell; Line 245: Correct “proteoglucans” to “proteoglycans”

Response 6: Thank you. All corrections related to minor changes have been made throughout the manuscript and highlighted in the revised manuscript.

Comments 7: Ensure that all references follow the same format.

Response 7: All references have been checked carefully and some of them have been completed (refs 6, 21, 45, 60, and 67).

Comments 8: Ensure that all abbreviations are clearly defined when first used in text. A list of abbreviations would be helpful.

Response 8: All abbreviations have clearly defined when used for the first time. A list is provided in separate sheet and could be optionally inserted into the manuscript.